# Bacterial Actin-Specific Endoproteases Grimelysin and Protealysin as Virulence Factors Contributing to the Invasive Activities of *Serratia*

**DOI:** 10.3390/ijms21114025

**Published:** 2020-06-04

**Authors:** Sofia Khaitlina, Ekaterina Bozhokina, Olga Tsaplina, Tatiana Efremova

**Affiliations:** Institute of Cytology, Russian Academy of Sciences, 194064 St. Petersburg, Russia; bozhokina@yahoo.com (E.B.); olga566@mail.ru (O.T.); tefi45@mail.ru (T.E.)

**Keywords:** actin proteolysis, metalloproteinases, protease ECP 32, grimelysin, protealysin, bacterial invasion

## Abstract

The article reviews the discovery, properties and functional activities of new bacterial enzymes, proteases grimelysin (ECP 32) of *Serratia grimesii* and protealysin of *Serratia proteamaculans*, characterized by both a highly specific “actinase” activity and their ability to stimulate bacterial invasion. Grimelysin cleaves the only polypeptide bond Gly42-Val43 in actin. This bond is not cleaved by any other proteases and leads to a reversible loss of actin polymerization. Similar properties were characteristic for another bacterial protease, protealysin. These properties made grimelysin and protealysin a unique tool to study the functional properties of actin. Furthermore, bacteria *Serratia*
*grimesii* and *Serratia proteamaculans*, producing grimelysin and protealysin, invade eukaryotic cells, and the recombinant *Escherichia coli* expressing the grimelysin or protealysins gene become invasive. Participation of the cellular c-Src and RhoA/ROCK signaling pathways in the invasion of eukaryotic cells by *S. grimesii* was shown, and involvement of E-cadherin in the invasion has been suggested. Moreover, membrane vesicles produced by *S. grimesii* were found to contain grimelysin, penetrate into eukaryotic cells and increase the invasion of bacteria into eukaryotic cells. These data indicate that the protease is a virulence factor, and actin can be a target for the protease upon its translocation into the host cell.

## 1. Introduction

Invasion of opportunistic bacterial pathogenic into eukaryotic cells is a process of interaction of bacteria with eukaryotic cells. [1,2]. To invade host cells, invasive bacteria should initiate the host cell signaling system by the proteins that interact with cell receptors and modify cytoskeleton through direct interaction with actin and actin binding proteins. Bacterial effectors can mimic natural activators of small GTPases or directly stimulate the host signaling pathways [3,4]. On the other hand, efficiency of bacterial invasion depends on the physiological state of host cells and is determined by the processes associated with changes in the distribution of cell surface receptors and cytoskeleton rearrangements [5]. This implies the presence of specific bacterial virulence factors capable of interacting with or modifying eukaryotic cell receptors, as well as components of the signal transduction system and actin cytoskeleton.

Studying the mechanisms of actin polymerization, we discovered and characterized a new enzyme-bacterial protease that exhibited highly specific “actinase” activity [6]. The protease turned out to be an intracellular enzyme of bacteria originally identified by standard morphological, biochemical, and cultural properties as an atypical lactose-negative strain of *Escherichia coli* A2 [7]. The actin-specific protease isolated from these bacteria was identified as a single 32 kDa polypeptide, which gave the protease the name ECP 32 (Escherichia coli protease, 32 kDa) [7,8]. Protease ECP 32 cleaves actin at a single site between Gly-42 and Val-43 within the DNase I-binding loop on the top of the actin monomer [9,10] that is involved in extensive interactions with the neighboring subunits within the actin filament [11,12]. The high specificity of actin proteolysis made ECP 32 a unique tool to study actin properties and interactions [13,14,15,16,17]. On the other hand, the enzyme by itself turned out to be not unique.

Although ECP 32 has been described as an *E. coli* A2 enzyme the *N*-terminal amino acid sequence of ECP 32 AKTSSAGVVIRDIF could not be identified in published *Escherichia coli* genomic sequences [8,18]. Therefore, the systematic position of the ECP 32-producing bacterial strain was reinvestigated. Using about 50 biochemical reactions of the Vitek-2 system (bioMeґrieux, Marcy l’Etoile, France) and partial sequencing of the 16S rRNA gene, the former *Escherichia coli* A2 strain was identified as *Serratia grimesii* A2 [18]. Then, the presumptive gene coding for the reference *S. grimesii* 30063 or *Serratia grimesii* A2 was cloned using published sequences of a similar protease protealysin identified in *S. proteamaculans* [19]. Amino acid sequences of two corresponding proteins from the *S. grimesii* A2 (former *E. coli* A2) and the reference strain of *S. grimesii* 30,063 were identical and contained the *N*-terminal 14 amino acids of protease ECP 32 as previously determined [8,18]. The same specific actin-hydrolyzing activity, characteristic of protease ECP 32 [9,10], was also revealed in bacterial extracts of the reference *S. grimesii* strain and *Escherichia coli* transformed by the presumptive gene encoding grimelysin (ECP32) in *S. grimesii* A2 [18]. Taken together, these data suggested that ECP 32 and grimelysin is the same enzyme named grimelysin [18], slightly different from protealysin [19]. While *S.grimesii* and *S. proteamaculans* belong to a cluster of bacteria within the *Serratia liquefaciens* group, a similar protease could be expected to be synthesized by other members of this group. Our preliminary data indicate that protease with the actinase activity is present in *Serratia marcescens*.

Along with the high similarity of the bacteria *S. grimesii* and *S. proteamaculans*, their proteases, grimelysin and protealysin, are highly homologous and differ by only 8 amino acid residues [18,19]. Moreover, similarly to the grimelysin-producing bacteria [20,21], *S. proteamaculans* 94 turned out to be one more bacterial strain where the actinase activity of metalloproteinase protealysin is coupled with bacterial invasion [21,22]. These results are consistent with the idea of the actin-specific metalloproteases being a factor that can be involved in bacterial invasion of eukaryotic cells. This paper describes the properties of grimelysin and protealysin in vitro and in vivo in the context of this idea. Our data indicate that the protease is a virulence factor and actin can be a target for the protease upon its translocation into the host cell.

## 2. Basic Properties and Substrate Specificity of Grimelysin and Protealysin

*Grimelysin (ECP 32)*, discovered, purified and initially characterized as protease ECP 32 [6,7,8], was later shown to be identical to grimelysin [18]. Therefore, the properties of the enzyme identified for ECP 32 could be applied to grimelysin. However, here we retain the name grimelysin (ECP 32) and ECP-cleaved actin to comply with the published data where the protease was named ECP 32. Grimelysin (ECP 32), purified from a bacterial extract using sequential chromatography steps, is a single 32 kDa polypeptide, whose *N*-terminal sequence was determined to be AKTSSAGVVIRDIFL [8]. The optimum of the protease activity was observed in the range of pH 7–8 when actin and melittin were used as substrates [8,23]. The proteolytic activity increased with increasing ionic strength: in 50–100 mM NaCl the activity of grimelysin (ECP 32) towards melittin was shown to be nearly twice higher than in a low ionic strength solution [23,24]. It was also enhanced in the presence of millimolar ATP concentrations, though hydrolysis of melittin was not accompanied by ATP hydrolysis at a rate comparable with the cleavage rate. This implies that protease grimelysin (ECP 32) is not an ATP-dependent enzyme [23], which is important for the experiments involving actin because actin contains ATP as a tightly-bound nucleotide. The protease activity is inhibited by EDTA, EGTA, o-phenanthroline and zincone, and the EDTA-inactivated enzyme can be reactivated by cobalt, nickel and zinc ions [2,3]. Based on these data, grimelysin (ECP 32) was classified as a neutral metalloproteinase (EC 3.4.24) [8].

Limited proteolysis of skeletal muscle actin between Gly-42 and Val-43 [10] was observed at enzyme: substrate mass ratios of 1:25 to 1:3000 [8]. Two more sites, between Ala-29 and Val-30 and between Ser-33 and Ile-34, were cleaved by ECP 32 in heat- or EDTA-inactivated actin, apparently due to conformational changes around residues 28–34 buried in intact actin [8]. Besides actin, only melittin [18,19], histone H5, bacterial DNA-binding protein HU and chaperone DnaK [25] were found to be protease substrates. In agreement with this high substrate specificity, ECP 32 did not hydrolyze tropomyosin, troponin, α-actinin, casein, histone H2B, ovalbumin, bovine serum IgG, bovine serum albumin, bovine pancreatic ribonuclease A, trypsin, human heat shock protein HSP70, chicken egg lysozyme [7], insulin [24], DNAse I [9,13], gelsolin [14] and profilin [26]. The amino acid residues recognized by grimelysin (ECP 32) in actin and melittin are hydrophobic. This specificity is characteristic for thermolysin-like metalloproteinases [27]. However, high specificity of the enzyme seems to be determined predominantly by conformation at the actin cleavage site rather than its primary structure.

*Grimelysin* was obtained as a recombinant protein. This has been achieved by cloning the putative gene encoding grimelysin in *S. grimesii* A2 and in the reference *S. grimesii* 30063 [18] using published protealysin sequences identified in *S. proteamaculans* [19]. Grimelysin shared all properties characteristic for ECP 32 including a molecular weight of 32 kDa, an *N*-terminal 14 amino acid sequence, optimum activity in the range of pH 7–8 and inhibition with o-phenanthroline and EGTA [18].

*Protealysin* is a neutral zink-containing metalloprotease of *Serratia proteamaculans*. The protealysin gene was cloned from a genomic library of *S. proteamaculans* strain 94 isolated from spoiled meat. This protein was expressed in *Escherichia coli* and purified as described earlier [19]. Similarly to other thermolysin-like proteases [27,28], protealysin is synthesized as a precursor containing a propeptide of about 50 amino acids that is removed during formation of mature active protein [29]. The propeptide is much shorter than the propeptides of the thermolysin-like proteases and has no significant structural similarity to the propeptides of most thermolysin-like proteases [30,31,32]. A similar propeptide of 50 amino acids was also detected in the primary structure of the recombinant grimelysin. According to SDS-electrophoresis, recombinant proteins with or without propeptide had an apparent molecular weight of 37 and 32 kDa, respectively [19].

The molecular weight of the active recombinant protealysin 32 kDa and the *N*-terminal amino acid sequence AKTSTGGEVI are identical to those of grimelysin [8,19]. The optimal pH for azocasein hydrolysis is 7, and protealysin is completely inhibited by *o*-phenanthroline [19], i.e., has the same properties as grimelysin [8,18]. Protealysin and grimelysin (ECP 32) are also similar in their unique property of being able to digest actin specifically [8,9,10,22,33].

## 3. Specific Actinase Activity of Grimelysin (ECP 32) and Protealysin

The ECP 32-like limited proteolytic activity towards actin appears in bacterial lysates of *S. grimesii* and *S. proteamaculans* only at the late stationary phase of bacterial growth [7,18,22]. The bacterial lysates as well as the purified grimelysin (ECP 32) or recombinant grimelysin and protealysin cleave actin at a single site giving rise to two fragments of 36 and 8 kDa (Figure 1A) [10,13,22]. The fragments remain associated in the presence of the tightly bound calcium or magnesium cation needed to maintain the native actin conformation and dissociate after removal of the tightly bound cation with EDTA [13]. The *N*-terminal sequence of the 36 kDa actin fragment produced by ECP 32 was determined as Val-Met-Val-Gly-Met [10]. The same sequence was determined for the 36 kDa *N*-terminal actin fragment produced by the lysates of recombinant bacteria expressing the grimelysin or protealysin gene [18,22]. According to the amino acid sequence of actin [34], this peptide corresponds to the cleavage site between Gly-42 and Val-43 [10], which is not attacked by any known proteases. However, in the case of protealysin, this cleavage pattern was observed only at the protealysin/actin mass ratio of 1:50 or lower. At a higher protealysin to actin ratio, the 36 kDa fragment was further cleaved yielding two closely situated bands with an apparent molecular weight of 33 kDa [22]. The *N*-terminal sequences of these fragments are Leu-Lys-Tyr-Pro-Ile-Glu and Ile-Leu-Thr-Leu-Lys-Tyr, corresponding to the cleavage of the bonds Thr66–Leu67 and Gly63–Ile64, respectively [22]. The difference in activity of the purified bacterial grimelysin and recombinant protealysin may be due to their origin, both in terms of different bacteria and different purification protocols. Therefore, comparison of the actinase activities of recombinant grimelysin and protealysin would be of interest.

## 4. Specific Properties of Protease-Cleaved Actin

Although actin cleaved with grimelysin (ECP 32) between Gly42 and Val43 preserves its native conformation [10,13], the cleaved actin completely loses its ability to polymerize in the presence of Ca^2+^ [5,8] (Figure 1B). The ability of the cleaved actin to polymerize is partially restored upon substitution of the tightly bound Ca^2+^ with Mg^2+^ [13]. However, the degree of this polymerization is lower than that of intact actin, while the critical concentration of polymerization of ECP-Mg-actin and the exchange of subunits in the ECP-Mg-actin polymer increase 30 and 20–30 times, respectively (Figure 1B) [13]. This effect is due to a more open conformation of the cleaved actin monomers, which decreases the monomer nucleation step of cleaved actin polymerization or/and increases the actin filament dynamics, i.e., the dissociation/association kinetics of actin monomers at the filament ends [15]. The open conformation of the ECP-cleaved actin monomers is also preserved in the ECP-actin polymer [15]. Similar properties are inherent to actin cleaved with protealysin [36]. Thus, cleavage in globular actin of the only amino acid bond between Gly-42 and Val-43 causes local conformational changes that weaken the intermonomer contacts during actin polymerization and lead to enhanced polymer dynamics. As it turned out later, these properties correlated with the ability of the protease-producing bacteria to invade eukaryotic cells [20,21,22] (Figure 1C)

Similar instability arises if the protease cleaves the bond between Gly-42 and Val-43 in the subunits of the actin polymer. Incubation of filamentous actin (F-actin) with purified *S. proteamaculans* protealysin or with the lysates of the recombinant *Escherichia coli* producing protealysin showed that 20–40% of F-actin was digested at enzyme/actin mass ratios rising from 1:50 up to 1:5, respectively [33]. This process was accompanied by the increased steady-state ATPase activity (dynamics) of F-actin [33], which was also shown to be characteristic of the polymers forming from grimelysin (ECP 32)-cleaved actin [13]. The cleavage-produced increase in the dynamics of the modified filaments can be reversed with phalloidin [13], aluminium and sodium fluorides [22,36], as well as with actin-binding proteins gelsolin [14], myosin subfragment 1 [37] and tropomyosin [17], indicating that actin-binding proteins can restore integrity of actin filaments damaged by proteolysis.

This assumption was verified in vivo using microinjection of the purified protease grimelysin (ECP 32) preparation into *Amoeba proteus* cells, which revealed dynamics of the cytoskeleton structures by alterations in amoeba locomotion [38]. After injection of the protease solution, pseudopodia formation was ceased, and the cytoplasm motility slowed down and finally stopped [38]. Injected amoebae remained spread and immobile until the locomotion was slowly restored to the control level. No changes in the locomotion were observed when protease grimelysin (ECP 32) was injected in the presence of ECP 32-specific antibodies [38]. These results indicate that the protease-produced modifications of actin cytoskeleton can be reversible and might be used by bacteria to regulate their intracellular activity within eukaryotic and probably bacterial cells. Specifically, actin-like, bacterial proteins that form the bacterial cytoskeleton could be protease substrates.

## 5. Actin-Like Proteins of Bacteria

Many functions of bacterial cells are performed by filamentous structures similar to those of the cytoskeleton filaments of eukaryotic cells [39,40]. These structures are formed by globular proteins, which are regarded as actin homologues, actin-like proteins or bacterial actins [39] because three-dimensional structures of their monomers are similar to that of skeletal actin [39,40]. Also similarly to the eukaryotic globular actin, globular bacterial actin-like proteins assemble into two-stranded filaments consisting of two protofilaments coming together in various ways [39,40], although this sets them apart from the eukaryotic actin filaments assembled only in the form of two parallel strands. The most studied bacterial cytoskeleton proteins, MreB, ParM, MamK and Ftz, perform such functions as shape determination in rod-shaped bacteria, cell division, plasmid segregation and organelle positioning [39,40]. It would be tempting, therefore, to suggest that the actin-specific proteases, grimelysin and protealysin, could regulate these activities. However, all these bacterial proteins, including MreB, most closely related to actin [40,41,42] differ from eukaryotic actin due to the absence of the DNase I-binding loop inserted in eukaryotic actin subdomain 2 [41,42], where eukaryotic actin is cleaved by grimelysin or protealysin. Thus, in contrast to eukaryotic actin, bacterial actin-like proteins do not contain any specific cleavage site to be attacked by grimelysin or protealysin. This suggests that the specific actinase activity of grimelysin and protealysin against eukaryotic actin could be important if the protease is delivered by bacteria into eukaryotic cells.

## 6. Invasive Activity of Bacteria *Serratia grimesii* and *Serratia proteamaculans*

Capability of the actinase-producing bacteria to invade eukaryotic cells was first found upon incubation of human larynx carcinoma Hep-2 cells with bacteria *Serratia grimesii* A2 (*E. coli* A2) [20,43]. After 2 h of infection, bacteria were detected within the cells, mostly in vacuoles but also free in cytoplasm (Figure 1C). The invasion was accompanied by a change in the shape of the eukaryotic cells, disappearance of the actin stress fibers inside the cells and appearance of protrusions on the cell surface [43]. No changes were observed in the cells infected with the wild-type *Escherichia coli* CCM 5172 that do not produce actinases [43]. Correlation between synthesis of the specific actin hydrolyzing protease and the ability of bacteria to invade eukaryotic cells was also detected in *Shigella flexneri* mutant obtained by exposure of bacteria to furazolidone [44,45]. This treatment resulted in the formation of *Shigella flexneri* L forms whose revertants were not pathogenic but able to invade eukaryotic cells, while their extracts are capable of cleaving actin as the bacterial extracts containing grimelysin or protealysin (Figure 2) [44,45].

It was also shown that the transformed cells are more sensitive to invasion by actinase-producing bacteria than non-transformed or poorly transformed cells [20,46]. Specifically, the grimelysin (ECP 32)-producing bacteria invaded the transformed epithelial and fibroblasts cells A431, HeLa and 3T3-SV40, but they were not found in embryonic fibroblasts, primary human keratinocytes and in cells of poorly transformed 3T3 cell lines [20]. Later, a quantitative analysis of invasion confirmed rather a low susceptibility of 3T3 cells to *Serratia grimesii* invasion, compared to that of 3T3-SV40 cells [46]. The higher susceptibility of the immortal CaCo-2 and HeLa cells compared to their untransformed counterparts was also shown for invasion of *Listeria monocytogenes* [47]. Moreover, transfection of resistant *Listeria monocytogenes* cells by SV40 large T antigen was shown to induce highly transformed continuous cell lines with a susceptibility to bacteria phenotypes [48]. This difference may be produced by the different set of cell surface receptors contacting with bacteria in the “normal” and transformed cells, as is shown, for example, for the G-protein-coupled receptors [49].

To find out if grimelysin and protealysin are actively involved in the entry of *Serratia grimesii* and *Serratia proteamaculans* into host cells, human larynx carcinoma Hep-2 cells were infected with recombinant *Escherichia coli* expressing grimelysin or protealysin gene [21]. The results of this work showed that the extracts of the recombinant bacteria cleave actin at the only site corresponding to the cleavage with grimelysin or protealysin [21]. Recombinant *Escherichia coli* carrying the grimelysin or protealysin gene were found in the eukaryotic cells, both in vacuoles (Figure 3) and free in cytoplasm [21]. At the same time, no invasion was observed if the Hep-2 cells were incubated with the non-invasive *Escherichia coli*-carrying plasmids that did not contain the protease gene. Internalization of the non-invasive *Escherichia coli* was also not observed if protealysin was added to the culture medium [21]. These results showed the direct participation of grimelysin and protealysin in the invasion of the host cells by the protease-producing bacteria [21].

Using electron microscopy, two modes of *Serratia grimesii* invasion were revealed. In most cases, interaction of *Serratia grimesii* with eukaryotic HeLa M cells starts with formation of a tight contact between bacteria and the host cell, which is followed by bacteria internalization, apparently due to a specific interaction of bacterial adhesins with a specific cell surface receptor (Figure 4A,B) [50]. This process corresponds to the “zipper” mechanism of invasion involving activation of the signal system of the host cell followed by moderate cytoskeleton rearrangements [51,52,53]. However, in some cases, bacteria looked like they were trapped by filopodia (Figure 4C,D), probably induced by injected bacterial proteins triggering the bacterial uptake, as described in the “trigger mechanism” of invasion [54,55,56,57]. Recently, coexistence of both the trigger and zipper invasion mechanisms was postulated for *Salmonella* invasion [58,59,60,61]. These data provide a frame for revealing bacterial and cellular factors involved in *Serratia grimesii* and *Serratia proteamaculans* invasion.

## 7. Bacterial Virulence Factors Involved in *Serratia* Invasion

Transformation of the non-invasive *Escherichia coli* with the protealysin gene led to the appearance in the bacteria of the invasive activity [21]. However, inactivation of this gene did not abolish the ability of *Serratia proteamaculans* to invade eukaryotic cells [62]. Moreover, invasive activity of *Serratia proteamaculans* was five times higher than that of *Serratia grimesii* [63]. This indicates involvement of other factors regulating invasion.

Bacteria of *Serratia* genus are known to be facultative pathogens able to induce nosocomial infections or infections in immunocompromised patients [64,65,66]. A most pathogenic strain of *Serratia* genus is *Serratia marcescens*, pathogenicity of which is determined by a secreted pore-forming toxin (hemolysin) ShlA [67,68] and extracellular proteases [69]. Probing of *Serratia grimesii* and *Serratia proteamaculans* for these factors revealed the presence and high activities of pore-forming hemolysin ShlA and extracellular metalloprotease serralysin in *Serratia proteamaculans.* At the same time, in *Serratia grimesii,* the activity of the toxin ShlA was not detected, and the serralysin activity of the bacterial growth medium was very low [63]. It was also shown that iron depletion strongly enhanced the invasive activity of *S. proteamaculans*, increasing activities of hemolysin ShlA and serralysin, but did not affect these properties of *Serratia grimesii* [63]. These results showed that, along with protealysin, the invasive activity of *S. proteamaculans* is also determined by hemolysin and serralysin [63]. At the same time, grimelysin remains the only known virulent factor of *S. grimesii*.

The first step in bacterial invasion is the contact of bacteria with the surface of eukaryotic cells, often performed by bacterial outer membrane protein OmpX [70,71,72]. Indeed, transformation of *Escherichia coli* by a plasmid carrying the OmpX gene of *Serratia proteamaculans* caused a 3-fold increase in the adhesion of bacteria to the surface of eukaryotic cells Hep G2 and DF-2, without producing any effect on *Escherichia coli* invasion [73]. On the other hand, our preliminary data indicate that OmpX is a substrate for protealysin capable of enhancing the invasion of *Serratia proteamaculans* carrying the inactivated protealysin gene [73]. These data indicate that along with its direct participation in bacterial adhesion, OmpX could be involved in bacterial invasion, while, in turn, protealysin could regulate bacterial adhesion.

## 8. Cellular Factors Involved in *Serratia* Invasion

The penetration of bacteria into eukaryotic cells includes the contact of bacteria with eukaryotic cells, activation of the cell signaling system, reorganization of the cytoskeleton leading to bacterial uptake and spreading of the invaded bacteria within and between the cells [55,74,75]. The efficiency of this process is determined not only by the activity of bacterial virulent factors, but also by the sensitivity of eukaryotic cells, which, in turn, correlates with the degree of their transformation [20,47] and depends on the cell environment [76,77]. These factors were employed to find out whether bacterial invasion is sensitive to the cell type and the presence of antioxidant *N*-acethylcysteine (NAC) in the culture medium. The results of these experiments confirmed different effects of NAC on 3T3, 3T3-SV40 and HeLa cells [78]. Incubation of HeLa cells with NAC increased penetration of grimelysin-producing bacteria by 1.5–2 times for wild-type *Serratia grimesii* and by 3–3.5 times for recombinant *Escherichia coli* expressing the grimelysin gene [79]. These effects did not correlate with the cytoskeleton rearrangements but might be due to the NAC-induced upregulation of cell surface receptors playing a primary role in cell adhesion and cell–cell junctions [79].

Specifically, one of the cell surface receptors whose expression is regulated by *N*-acethylcysteine is E-cadherin, a transmembrane receptor known to be upregulated by NAC [77] and one of the two cell surface proteins that mediate adhesion and internalization of *Listeria monocytogenes* within epithelial cells [80,81]. Binding of bacteria internalin InIA to E-cadherin triggers the c-Src kinase-mediated phosphorylation of E-cadherin, followed by ubiquitination and E-cadherin internalization via the clathrin-mediated pathway [80,81,82]. In line with these data, incubation of 3T3 and 3T3-SV40 cells with NAC led both to increased expression of E-cadherin and increased sensitivity of these cells to invasion [46]. Co-localization of *S. grimesii* with E-cadherin of the 3T3 and 3T3-SV40 cells was also shown [46]. In addition, inhibitory analysis with the ROCK and c-Src kinase inhibitors revealed a correlation between c-Src and ROCK protein kinase activities, expression of E-cadherin and invasive activity of *Serratia grimesii* [83]. These data allow us to suggest that E-cadherin is at least one of the receptors involved in *Serratia grimesii* invasion.

On the other hand, participation of OmpX in *Serratia proteamaculans* invasion indicates that other cell surface receptors, including integrins and the epidermal growth factor receptor (EGFR), could be involved in this process, as it has been recently shown for the invasion of *Salmonella* [59,60,61]. In line with these data, infection of epithelial M-HeLa cells by bacteria *Serratia proteamaculans* led to the changes in localization of EGFR and fibronectin receptors α5-, β1-integrins. Accumulation of α5-, β1-integrins on the cell surface was accompanied by intensive attachment of bacteria to the sites of α5-integrin localization [84], indicating involvement of the outer membrane protein OmpX–fibronectin–integrin in the penetration of *Serratia proteamaculans* into eukaryotic cells.

## 9. Conclusions

Though *Serratia* are facultative pathogens able to cause nosocomial infections or infections in immunocompromised patients, hospital infection by *S. grimesii* or *S. proteamaculans* is low [64,65,66]. Consistent with this, the capability of *S. grimesii* to invade eukaryotic cells is also rather low, with only about 10% of the cells being invaded either by the wild-type or recombinant bacteria when cultured cells are infected in vitro [21]. This implies that production of grimelysin or protealysin does not make the bacteria pathogenic but rather provides them with an opportunity to be rendered pathogenic under specific conditions. Therefore, *Serratia grimesii* and *Serratia proteamaculans* invasion combines specific bacterial mechanisms to contact eukaryotic cells and the specificity of the cells to be invaded.

To penetrate eukaryotic cells, invasive bacteria should initiate their uptake by activating the signal transduction system and cytoskeleton rearrangements [1,2,3,4]. Thereby, pathogenic bacteria use specialized secretion systems to deliver bacterial virulence factors directly into the cytoplasm of the host cell to manipulate the intracellular pathways, thus following the trigger invasion mechanism [53,54,55,56,57,85]. In contrast, facultative bacterial pathogens use specific surface proteins that interact with the host cell receptors, according to the zipper invasion mechanism. These interactions activate receptor-mediated cell signaling cascades and lead to bacterial uptake [2,4,60,61,86]. Recent data showed however that the virulence components transported via the type III secretion system of pathogenic bacteria can be also translocated into the eukaryotic cells by the bacterial outer membrane vesicles (OMVs) [87]. Importantly, OMVs can be produced by opportunistic bacteria as well as by pathogenic ones [88,89,90,91,92], thus smoothing the difference between pathogenic bacteria and facultative bacterial pathogens.

Penetration of opportunistic bacteria *Serratia grimesii* and *S. proteamaculans* into eukaryotic cells [21] correlates with the presence in these bacteria of homologous metalloproteases grimelysin and protealysin, respectively [18,22]. Protealysin was found in the cytoplasm of 3T3-SV40 cells infected with *S. proteamaculans* or recombinant *Escherichia coli* expressing the protealysin gene [33], suggesting that protease can be transported into the cytoplasm of eukaryotic cells. Consistent with this suggestion, our preliminary data showed that the outer membrane vesicles produced by *Serratia grimesii* were able to transfer protease grimelysin into the cytoplasm of eukaryotic cells. This transfer enhanced the penetration of bacteria into eukaryotic cells if these cells were incubated with the vesicles prior to infection [93,94]. At least a part of this effect could be produced by the involvement of the OMV-delivered protease in the cytoskeleton dynamics. We have previously shown that the cleavage of actin with grimelysin (ECP32) enhanced turnover of actin monomers within actin filaments [15,38]. This, in turn, could contribute to the rearrangements of the actin cytoskeleton leading to bacteria internalization. The invasive activity of *Serratia* can also be regulated by activities of the intracellular factors, being, for example, diminished upon transfection of eukaryotic cells with anti-RhoA siRNA [83]. The cells transfected with anti-RhoA siRNA exhibited cell rounding, disassembly of actin cytoskeleton and formation of protrusions at the cell periphery [83], indicating an active involvement of the cytoskeleton in *S. grimesii* uptake.

Thus, the transition of opportunistic bacteria to the status of pathogenic is associated both with the activity of bacterial virulence factors and the sensitivity of eukaryotic cells to bacteria. In particular, the sensitivity of eukaryotic cells to bacterial invasion correlates with the degree of their transformation [20,46,47], the effects of antioxidants [78,79] and the activity of cell signaling components like c-SRC and ROCK kinases [83]. Further work is needed to integrate these individual cellular factors and bacterial virulence factors into the *Serratia* invasion mechanism.

## Figures and Tables

**Figure 1 ijms-21-04025-f001:**
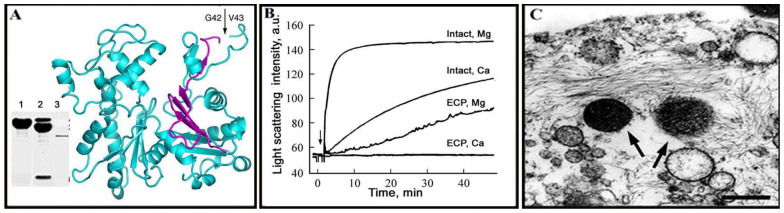
The actinase activity of grimelysin (ECP 32). (**A**) Three-dimensional structure of skeletal muscle actin cleaved with grimelysin (ECP 32) at a single site, Gly42-Val43, within the DNase I-binding loop [35]. Inset shows SDS electrophoresis of actin (lane 1), with the actin 36 and 8 kDa fragments produced by the grimelysin (ECP32) cleavage; the upper band is actin not completely cleaved in this experiment (lane 2) and isolated grimelysin (ECP 32) (lane 3) [8,10]. (**B**) Polymerization of ECP-cleaved skeletal muscle actin compared to that of intact actin. (**C**) Invasion of human larynx carcinoma Hep-2 cells by *Serratia grimesii* 30063. Intracellular bacteria are marked with arrows. Scale bar, 1 µm. Arrows indicate intracellular bacteria Reproduced from [13,21,35] with permissions from Elsevier Licences 4820990651384 and 1032953-1, 2020 (**A**,**B**) and from Wiley Licence 4821290007049, 2020 (**C**).

**Figure 2 ijms-21-04025-f002:**
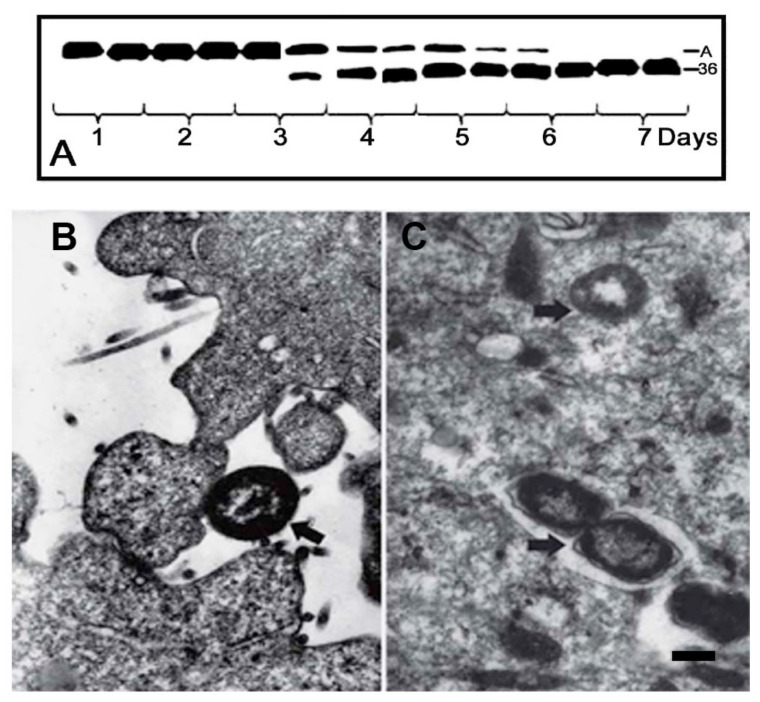
Apathogenic *Shigella flexneri* 5a2c mutant treated with furazolidone can invade eukaryotic cells. (**A**) Detection of the actinase activity in the lysates of *Shigella flexneri* 5a2c acquired upon the furazolidone treatment. Numbers indicate the time points of bacteria growth when their extracts were tested for the actinase activity. A, actin; 36, the 36 kDa actin fragment. (**B**) Incubation of human larynx carcinoma Hep-2 cells with the furazolidone-treated non-pathogenic *Shigella flexneri* 2a 4115. (**C**) Invasion of the pathogenic furazolidon-treated *Shigella flexneri* 5a2c into human larynx carcinoma Hep-2 cells [39]. Arrows indicate extracellular bacterium (**B**) and intracellular bacteria lying in vacuoles (**C**). Scale bar, 0.5 µm. Reproduced from [45] with permission from Springer Nature License 4823180887079, 2020.

**Figure 3 ijms-21-04025-f003:**
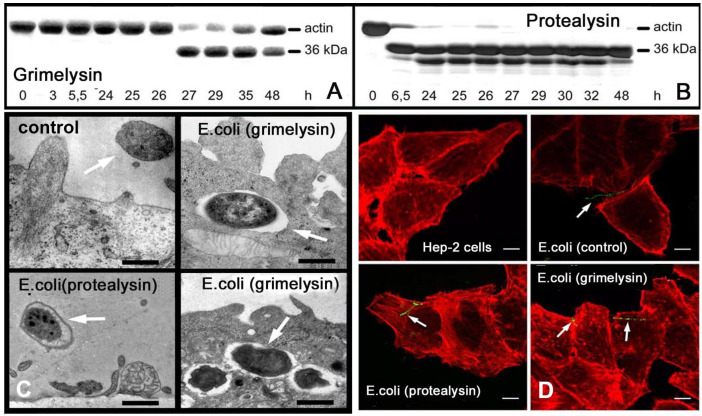
Actinase activity of recombinant *E. coli* transformed with the grimelysin or protealysin gene. (**A**,**B**) Detection of the actinase activity in the lysates of the recombinant *E. coli* expressing grimelysin or protealysin. Numbers indicate the time points of bacteria culturing when their extracts were tested for actinase activity [21]. (**C**,**D**) Invasion of human larynx carcinoma Hep-2 cells by recombinant *E. coli* expressing the grimelysin or protealysin gene, observed by electron (**C**) and confocal (**D**) microscopy [21]. Arrows indicate extracellular and internalized bacteria. By electron microscopy, bacteria are in vacuoles (**C**) that are not visible with confocal microscopy. Scale bars, 1 µm. (**D**) The samples were examined under a Leica TCS SL confocal scanning microscope using a dual argon ion (488 nm; green fluorescence) and helium/neon (543 nm; red fluorescence) laser system to visualize the FITC-stained bacteria and rhodamine phalloidin stained cytoskeleton, respectively. Scale bars, 1 µm (**C**) and 10 µm (**D**). Modified from [21] with permission from Wiley license 4827670496420, 2020.

**Figure 4 ijms-21-04025-f004:**
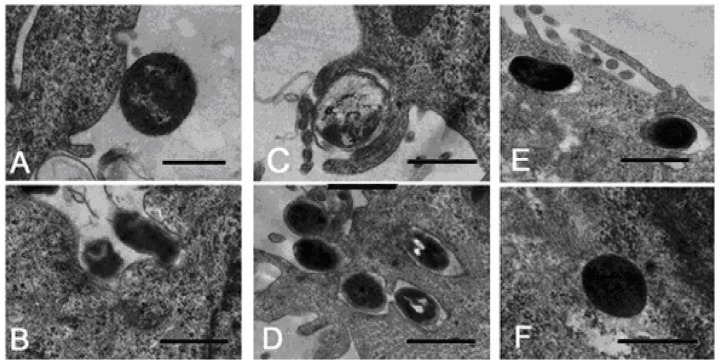
Electron microscopic identification of the initial steps in the invasion of eukaryotic cells by *Serratia grimesii*. (**A**) Tight contact of bacteria with HeLa M cells. (**B**) Extensive contact of bacteria with the surface of the host cell followed by the entry of bacteria into the host cell, which corresponds to the zipper mechanism of invasion. (**C**,**D**) Appearance of host cell filopodia, whose fusing traps bacterium and places it inside the vacuole, which corresponds to the trigger mechanism. (**E**) Bacteria live in the vacuoles. (**F**) Bacteria leave the vacuole to live free in the host cell cytoplasm. Scale bars, 1 µm. (Modified from [50] with permission from Pleiades Publishing, Ltd. 117342, 2020.).

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
