# Peer review of "Bacterial Actin-Specific Endoproteases Grimelysin and Protealysin as Virulence Factors Contributing to the Invasive Activities of Serratia"

_ijms, 2020, doi:10.3390/ijms21114025_

Round 1

Reviewer 1 Report

This is an interesting review article adressing both, functional modifications of actin as induced by limited proteolysis with specific bacterial proteases, and the mechanisms of invasion of certain bacteria into eucaryotic cells. It describes the efforts of research groups in thiese fields over several decades.

I have the following comments and suggestions which the authors should consider:

  1. The initial historical part of the introduction is relative verbose and lengthy. This part can be shortened down to a few sentences without affecting the scientific content of the paper.
  2. The terminology of the protease is somewhat confusing. It was initially called ECP32, referring to E.coli as which the bacterium was initially (but falsely) classified. Later it turned out that the source was in fact Serratia grimesii, and the term grimelysin was created. This should be explained once in the beginning of the article, and after that only the term grimelysin shoud be used throughout the text. The simultaneous use of both terms confuses the reader rather than doing anything good.
  3. The meaning of the sentence starting in line 190 with „ These results indicate…“ is not quite clear to me. I assume that it has something to do with the following chapter. Please rephrase the sentence to make more clear what you mean.
  4. The legend to Fig. 2 C,D needs some amendments, especially since these two figures are not referred to in the text of the paper. Where do the arrows point to? What is FITS?. A more detailed description is necessary.
  5. There are several other bacterial species known which invade eukaryotic cells, possibly by using different mechanisms. Since the article is a review I suggest to include a short chapter mentioning what is known about i.e. Listeria or Salmonella invasion methods. Both cases are mentioned in the text very sparsely, so it may be useful to add some more information, just for comparative purposes .
  6. Some additional sentences should be included – preferrably in the conclusions chapter - about the possible activity of grimelysin (on G-actin, F-actin or both) in the cytoplasm of the eukaryotic cells, considering the influence of the many actin binding proteins present. Also the authors conclude from their experiments that grimelysin and related proteins are one – possibly important - factor which enables entry of bacteria into the cytoplasm of eukaryotic cells. They should be more explicit in describing possible mechanisms by which this could be achieved, even if it requires some speculations.

Author Response

This is an interesting review article adressing both, functional modifications of actin as induced by
limited proteolysis with specific bacterial proteases, and the mechanisms of invasion of certain bacteria
into eucaryotic cells. It describes the efforts of research groups in thiese fields over several decades.

I have the following comments and suggestions which the authors should consider:
1. The initial historical part of the introduction is relative verbose and lengthy. This part can be
shortened down to a few sentences without affecting the scientific content of the paper.

Answer: As the paper has been prepared for a separate rather than a regular issue of the journal, the beginning
of the paper was aimed at reproducing the atmosphere of discovery which certainly does not happen every day.
Anyway, now we have modified the introduction in a scientific style. The initial story is removed and replaced by
suitable information. The introduction is edited, shortened and corrected.

2. The terminology of the protease is somewhat confusing. It was initially called ECP32, referring to
E.coli as which the bacterium was initially (but falsely) classified. Later it turned out that the
source was in fact Serratia grimesii, and the term grimelysin was created. This should be
explained once in the beginning of the article, and after that only the term grimelysin shoud be
used throughout the text. The simultaneous use of both terms confuses the reader rather than
doing anything good.

Answer: We certainly agree that it would be good to describe the situation in the beginning of the article and
use the term grimelysin throughout the text. However, all the actin experiments had been performed with ECP32
and the renaming would contrast to the original papers quoted, both in their titles and content. Therefore we
describe the situation in the introduction part of the article and save the name ECP32 in brackets as “grimelysin
(ECP 32)” as long as ECP32 is mentioned in the text. In the text that is not connected with experiments only the
term grimelysin is used.

3. The meaning of the sentence starting in line 190 with „These results indicate…“ is not quite clear
to me. I assume that it has something to do with the following chapter. Please rephrase the
sentence to make more clear what you mean.

Answer: The end of the paragraph is modified: These results indicate that the protease-produced modifications
of actin cytoskeleton can be reversible and might be used by bacteria to regulate their intracellular activity within
bacterial and eukaryotic cells.

4. The legend to Fig. 2 C,D needs some amendments, especially since these two figures are not
referred to in the text of the paper. Where do the arrows point to? What is FITS?. A more detailed
description is necessary.

Answer: I have to apologize! The numbers and positions of Figs.2 and Fig.3 had been mixed up in the
manuscript. The figures and legends to the figures were OK, but their numbers and positions in the text should be
the opposite. This produced an impression that these two figures are not referred to in the text of the paper. Now
everything is corrected and some additional information has been added if necessary. FITS is FITC, of course. The
legend to Fig.3 is complemented with more detailed and additional information. Bar values are added everywhere.

There are several other bacterial species known which invade eukaryotic cells, possibly by using
different mechanisms. Since the article is a review I suggest to include a short chapter mentioning
what is known about i.e. Listeria or Salmonella invasion methods. Both cases are mentioned in the
text very sparsely, so it may be useful to add some more information, just for comparative purposes.

Answer: Yes, a description of the mechanisms used by Listeria and Salmonella to invade eukaryotic cells would
be absolutely necessary if we could discuss the mechanisms of Serratia invasion. Unfortunately, our results that
would allow us at least to speculate about these mechanisms are not published yet. Therefore this article is still the
story about an actin-specific protease that – to our luck – is involved in bacterial invasion by a still unknown
mechanism. Nevertheless, while we have modified Introduction and Conclusions, we have added numerous
references to the article reviewing invasion mechanisms and bacterial virulence factors.

5. Some additional sentences should be included – preferably in the conclusions chapter - about the
possible activity of grimelysin (on G-actin, F-actin or both) in the cytoplasm of the eukaryotic
cells, considering the influence of the many actin binding proteins present. Also, the authors
conclude from their experiments that grimelysin and related proteins are one – possibly
important - factor which enables entry of bacteria into the cytoplasm of eukaryotic cells. They
should be more explicit in describing possible mechanisms by which this could be achieved, even
if it requires some speculations.

Answer: In the conclusions chapter, some mechanisms of a possible protease activity against cytoskeleton
reorganizations are now mentioned. So far we cannot be more explicit (see above).

Reviewer 2 Report

The paper titled „Bacterial actin-specific endoproteases grimelysin and 2 protealysin as virulence factors contributing to the 3 invasive activities of Serratiais a detailed review concerning grimelysin and protealysin enzymes. The biochemical properties of proteins have been thoroughly described, as well as the involvement of these proteases in bacterial invasion of eukaryotic cells. This review may be of high interest for researchers interested in Serratia virulence. However, I have some major comments on this paper.

For me, this paper is not well organized and the order of the presented information is not logical. Because of this, the article is not so easy to read and follow. There is a lack of information introducing the reader to the research, such as the characteristics of S. grimesii and proteamaculans, which will show why research on these pathogens is important. Section 6 (Bacterial virulence factors involved in the Serratia invasion) should be at the beginning of the paper to introduce the reader to the topic. First, information about Serratia (including S. grimesii and S. proteamaculans- why are they important from the medical point of view) should be presented, then what are the known mechanism of pathogenicity of these bacteria. And finally how grimelysin and protealysin fit in this topic. In the actual version first we learn a lot about the properties of a protein without knowing its role. In short, there is no introduction saying why the described bacteria and proteins are interesting. It would also be nice if the Authors could mention the homologs of grimelysin in other bacteria, like protease Prt1 in P. carotovorum.

The beginning of the introduction section is written in a popular science style. For example, I am not convinced that the phrase "it was a disaster" would fit the scientific publication. I do not mind this, but I think that for most scientists this style will not fit. Besides, this introduction do not give revelant informations. The article would not lose value if these informations were omitted.

I am not a native speaker but for me the English language and style need improvement. I have problems understanding what the Authors meant in many parts of the publication. Just for example lines 61-63, 108-110.

Figure 1: in lane 2 there are 3 bands, not just 36 and 8 kDa fragments

Since it is known that the strain E. coli A2 is S. grimesii, the Authors should continue to use the name Serratia, possibly S. grimesii A2 (E. coli A2) instead of E. coli A2 (S. grimesii)

In the abstract Authors mentioned the RhoA/ROCK and c-Src signaling pathways, but in the article, there is no information about RhoA. Moreover, the ROCK and Src roles should be clarified for readers not involved in this topic. Also, the Authors write once c-Src, and once Src. It should be standardized or the Authors should provide ab information that it is another name for the same kinase.

Section 4 (Actin-like proteins of bacteria) is unnecessary. I do not see the point of describing bacterial actin-like proteins if at the end of this section Authors provide the information that these proteins do not contain specific cleavage sites for grimelysin and protealysin. That one sentence would be enough. Moreover, the sentence in lines 209-211 does not make sense. Of course, that actinase activity against eukaryotic actin is important after delivery of bacteria to eukaryotic cells, because the actin is located in the cells.

Not all bacterial species names are written in italics.

Author Response

Reviewer: The paper titled „Bacterial actin-specific endoproteases grimelysin and 2 protealysin as
virulence factors contributing to the 3 invasive activities of Serratia” is a detailed review concerning
grimelysin and protealysin enzymes. The biochemical properties of proteins have been thoroughly
described, as well as the involvement of these proteases in bacterial invasion of eukaryotic cells. This
review may be of high interest for researchers interested in Serratia virulence. However, I have some
major comments on this paper.
For me, this paper is not well organized and the order of the presented information is not logical. Because
of this, the article is not so easy to read and follow. There is a lack of information introducing the reader
to the research, such as the characteristics of S. grimesii and proteamaculans, which will show why
research on these pathogens is important. Section 6 (Bacterial virulence factors involved in the Serratia
invasion) should be at the beginning of the paper to introduce the reader to the topic. First, information
about Serratia (including S. grimesii and S. proteamaculans- why are they important from the medical
point of view) should be presented, then what are the known mechanism of pathogenicity of these
bacteria. And finally how grimelysin and protealysin fit in this topic. In the actual version first we learn a
lot about the properties of a protein without knowing its role. In short, there is no introduction saying
why the described bacteria and proteins are interesting. It would also be nice if the Authors could
mention the homologs of grimelysin in other bacteria, like protease Prt1 in P. carotovorum.

Answer: We agree that the paper written by a suggested plan would be good to describe the role of S.
grimesii and S. proteamaculans in bacterial invasion if these bacteria would be an important part of the
hospital infection, their properties would be widel studied, and the authors would be medical scientists.
Such papers do exist but they are mostly concentrated on S.marcescens, as a realy pathogenic Serratia
species [Mahlen, S.D. Serratia infections: from military experiments to current practice. Clin. Microbiol.
Rev. 2011, 24, 755–791. DOI:10.1128/CMR.00017-11; Hertle, R.; Schwarz, H, Serratia marcescens
internalization and replication in human bladder epithelial cells. BMC Infect. Dis. 2004, 4, 16, DOI:
10.1186/1471-2334-4-16. Hertle, R. The family of Serratia type pore forming toxins. Curr. Protein Pept. Sci.
2005, 6, 313–325. DOI: 10.2174/1389203054546370]. As a hospitals infection, Serratia are facultative
pathogens able to cause nosocomial infections or infections in immunocompromised patients, and these
properties are studied rather scarily. We are not medical scientists. Moreover, invasion of S.grimesii is
shown to be rather low, with only about 10% of the cells being invaded either by the wild-type or
recombinant bacteria when cultures cells are infected in vitro [Bozhokina et al., 2011]. Hospital infection
with S. grimesii or S.proteamaculansis is rather low. This implies that production of the enzyme does not
make the bacteria pathogenic but rather provides them with an opportunity to render pathogenic under
specific conditions. Therefore the aim of our review paper is quite different from the medical aspects. We
have found that proteases grimelysin and protealysin isolated from S.grimesii and S.proteamaculans
specifically cleave actin at a site strongly involved in actin functional activity and is not cleaved by any
other protease. As cellular actin cytoskeleton is actively involved in the mechanisms of bacterial invasion,
we believe that grimelysin and protealysin can play an important role in this process. As these processes
are known rather poorly, our paper starts with the “actinase” properties of these new proteases probably
important for the invasion mechanisms rather that to start with the bacterial invasion. We have modified
Introduction and Conclusions of the paper to make them more clear.

Protease Prt1 from P. carotovorum is an extracellular protease, 69% homologous to protealysin of S
proteamaculans. This is rather far from the intracellular proteases that specifically cleave actin. Although
it would be interesting to identify proteases homologous to grimelysin/protealysin and evaluate their
anti-actin properties and the invasive potential of the corresponding bacteria, this is a separate work.
Reviewer: The beginning of the introduction section is written in a popular science style. For example, I
am not convinced that the phrase "it was a disaster" would fit the scientific publication. I do not mind
this, but I think that for most scientists this style will not fit. Besides, this introduction do not give
revelant informations. The article would not lose value if these informations were omitted.

Answer: As the paper has been prepared for a separate rather than a regular issue of the journal, the
beginning of the paper was aimed at reproducing the atmosphere of our discovery. We have modified
the introduction in a scientific style. The initial story is removed and replaced by suitable information.
Introduction is edited, shortened and corrected.

Reviewer: I am not a native speaker but for me the English language and style need improvement. I have
problems understanding what the Authors meant in many parts of the publication. Just for example lines
61-63, 108-110.

Answer: In response to the reviewer's recommendations, the text of the article has been read and edited
by an expert.
Lines 61-63 and 108-110 are modified.
Lines 61-63 (now 57-59): The same specific actin-hydrolyzing activity, characteristic of protease ECP 32 [9,
10], was also revealed in bacterial extracts of both the reference S. grimesii strain and the corresponding
recombinant protease from S. grimesii A2.
Lines 108-110 (now 103-105): Grimelysin was obtained as a recombinant protein. This have been achieved
by cloning the putative gene encoding grimelysin in S. grimesii A2 (E.coli A2) and in the reference S.
grimesii 30063 [18] using published protealysin sequences identified in S. proteamaculans [19].

Reviewer: Figure 1: in lane 2 there are 3 bands, not just 36 and 8 kDa fragments

Answer: The upper band in lane 2 is actin, which was not completely cleaved in this experiment. This
explanation is added to the figure legend.
Since it is known that the strain E. coli A2 is S. grimesii, the Authors should continue to use the name
Serratia, possibly S. grimesii A2 (E. coli A2) instead of E. coli A2 (S. grimesii)
This is corrected.

Reviewer: In the abstract Authors mentioned the RhoA/ROCK and c-Src signaling pathways, but in the
article, there is no information about RhoA. Moreover, the ROCK and Src roles should be clarified for
readers not involved in this topic. Also, the Authors write once c-Src, and once Src. It should be
standardized or the Authors should provide ab information that it is another name for the same kinase.

Answer: Information about RhoA inhibition is added to the text.
Several word about signaling role of ROCK and c-Src are added
Src is corrected for c-Src everywhere

Reviewer: Section 4 (Actin-like proteins of bacteria) is unnecessary. I do not see the point of describing
bacterial actin-like proteins if at the end of this section Authors provide the information that these
proteins do not contain specific cleavage sites for grimelysin and protealysin. That one sentence would be
enough. Moreover, the sentence in lines 209-211 does not make sense. Of course, that actinase activity
against eukaryotic actin is important after delivery of bacteria to eukaryotic cells, because the actin is
located in the cells.

Answer: Last sentence of the previous section is modified to justify this section. But, n any case, we
consider this section important. Bacterial proteases should have bacterial targets, and the actinase activity
against bacterial actin might be important for bacteria if their actin-like proteins would contain an
appropriate site. Therefore we think that this part of the text is important. As to the sentence in lines 209-
211, we have added several words to a previous sentence to remind that bacterial actin-like proteins
differ from eukaryotic actin by the absence of the DNase I-binding loop inserted in eukaryotic actin
subdomain 2 [41,42] where eukaryotic actin is cleaved by grimelysin or protealysin.

Reviewer: Not all bacterial species names are written in italics.

Answer: Corrected
We are grateful to the reviewer for useful comments and for discussing our work in terms of invasive
activity of bacteria. We will return to these recommendations when we learn more about the mechanisms
used by S. grimesii and S. proteamaculans to invade eukaryotic cells.

Round 2

Reviewer 2 Report

The Authors improved their paper in some aspects. For me the organization of work is still not logical, however, this is the vision of the Authors, we do not have to agree on this. Nevertheless, this article provides important information about grimelysin and protealysin enzymes that may be of interest to other scientists and therefore it would be a shame if the paper was not published. Regardless of the organization, I have minor comments:

  1. Reviewer:

 The paper titled „Bacterial actin-specific endoproteases grimelysin and 2 protealysin as virulence factors contributing to the 3 invasive activities of Serratia” is a detailed review concerning grimelysin and protealysin enzymes. The biochemical properties of proteins have been thoroughly described, as well as the involvement of these proteases in bacterial invasion of eukaryotic cells. This review may be of high interest for researchers interested in Serratia virulence. However, I have some major comments on this paper. For me, this paper is not well organized and the order of the presented information is not logical. Because of this, the article is not so easy to read and follow. There is a lack of information introducing the reader to the research, such as the characteristics of S. grimesii and proteamaculans, which will show why research on these pathogens is important. Section 6 (Bacterial virulence factors involved in the Serratia invasion) should be at the beginning of the paper to introduce the reader to the topic. First, information about Serratia (including S. grimesii and S. proteamaculans- why are they important from the medical point of view) should be presented, then what are the known mechanism of pathogenicity of these bacteria. And finally how grimelysin and protealysin fit in this topic. In the actual version first we learn a lot about the properties of a protein without knowing its role. In short, there is no introduction saying why the described bacteria and proteins are interesting. It would also be nice if the Authors could mention the homologs of grimelysin in other bacteria, like protease Prt1 in P. carotovorum.

Answer:

We agree that the paper written by a suggested plan would be good to describe the role of S. grimesii and S. proteamaculans in bacterial invasion if these bacteria would be an important part of the hospital infection, their properties would be widel studied, and the authors would be medical scientists. Such papers do exist but they are mostly concentrated on S.marcescens, as a realy pathogenic Serratia species [Mahlen, S.D. Serratia infections: from military experiments to current practice. Clin. Microbiol. Rev. 2011, 24, 755–791. DOI:10.1128/CMR.00017-11; Hertle, R.; Schwarz, H, Serratia marcescens internalization and replication in human bladder epithelial cells. BMC Infect. Dis. 2004, 4, 16, DOI: 10.1186/1471-2334-4-16. Hertle, R. The family of Serratia type pore forming toxins. Curr. Protein Pept. Sci. 2005, 6, 313–325. DOI: 10.2174/1389203054546370]. As a hospitals infection, Serratia are facultative pathogens able to cause nosocomial infections or infections in immunocompromised patients, and these properties are studied rather scarily. We are not medical scientists. Moreover, invasion of S.grimesii is shown to be rather low, with only about 10% of the cells being invaded either by the wild-type or recombinant bacteria when cultures cells are infected in vitro [Bozhokina et al., 2011]. Hospital infection with S. grimesii or S.proteamaculansis is rather low. This implies that production of the enzyme does not make the bacteria pathogenic but rather provides them with an opportunity to render pathogenic under specific conditions. Therefore the aim of our review paper is quite different from the medical aspects. We have found that proteases grimelysin and protealysin isolated from S.grimesii and S.proteamaculans specifically cleave actin at a site strongly involved in actin functional activity and is not cleaved by any other protease. As cellular actin cytoskeleton is actively involved in the mechanisms of bacterial invasion, we believe that grimelysin and protealysin can play an important role in this process. As these processes are known rather poorly, our paper starts with the “actinase” properties of these new proteases probably important for the invasion mechanisms rather that to start with the bacterial invasion. We have modified Introduction and Conclusions of the paper to make them more clear.

I understand this is not a medical review and the mechanism of invasion in the case of S. grimesii and S. proteamaculansis is still rather unknown. However, I think that at least basic information about these bacteria species would be valuable for this paper. The authors are co-authors of the majority of all papers concerning grimelysin form Serratia, but other researchers do not necessarily know that these bacterial species are poorly characterized. Just for example the modified information that the Authors provided in answers could be incorporated in the text- “Moreover, invasion of S.grimesii is shown to be rather low, with only about 10% of the cells being invaded either by the wild-type or recombinant bacteria when cultures cells are infected in vitro [Bozhokina et al., 2011]. Hospital infection with S. grimesii or S.proteamaculansis is rather low.”.

  1. lines: 58-61- the sentence is still incomprehensible: “The same specific actin-hydrolyzing activity, characteristic of protease ECP 32 [9,10], was also revealed in bacterial extracts of both the reference S. grimesii strain and the corresponding recombinant protease from S. grimesii A2.” If the activity was presented in bacterial extracts of both strains then “the sentence and the corresponding recombinant protease from” should be deleted. Unless the Authors meant that in the bacterial extract from S. grimesii and in the protease S. grimessi A2, then the word both is misleading.

Author Response

Comments and Suggestions for Authors

The Authors improved their paper in some aspects. For me the organization of work is still not logical, however, this is the vision of the Authors, we do not have to agree on this. Nevertheless, this article provides important information about grimelysin and protealysin enzymes that may be of interest to other scientists and therefore it would be a shame if the paper was not published. Regardless of the organization, I have minor comments:

  1. Reviewer:

 The paper titled „Bacterial actin-specific endoproteases grimelysin and 2 protealysin as virulence factors contributing to the 3 invasive activities of Serratia” is a detailed review concerning grimelysin and protealysin enzymes. The biochemical properties of proteins have been thoroughly described, as well as the involvement of these proteases in bacterial invasion of eukaryotic cells. This review may be of high interest for researchers interested in Serratia virulence. However, I have some major comments on this paper. For me, this paper is not well organized and the order of the presented information is not logical. Because of this, the article is not so easy to read and follow. There is a lack of information introducing the reader to the research, such as the characteristics of S. grimesii and proteamaculans, which will show why research on these pathogens is important. Section 6 (Bacterial virulence factors involved in the Serratia invasion) should be at the beginning of the paper to introduce the reader to the topic. First, information about Serratia (including S. grimesii and S. proteamaculans- why are they important from the medical point of view) should be presented, then what are the known mechanism of pathogenicity of these bacteria. And finally how grimelysin and protealysin fit in this topic. In the actual version first we learn a lot about the properties of a protein without knowing its role. In short, there is no introduction saying why the described bacteria and proteins are interesting. It would also be nice if the Authors could mention the homologs of grimelysin in other bacteria, like protease Prt1 in P. carotovorum.

Answer:

We agree that the paper written by a suggested plan would be good to describe the role of S. grimesii and S. proteamaculans in bacterial invasion if these bacteria would be an important part of the hospital infection, their properties would be widely studied, and the authors would be medical scientists. Such papers do exist but they are mostly concentrated on S.marcescens, as a really pathogenic Serratia species [Mahlen, S.D. Serratia infections: from military experiments to current practice. Clin. Microbiol. Rev. 2011, 24, 755–791. DOI:10.1128/CMR.00017-11; Hertle, R.; Schwarz, H, Serratia marcescens internalization and replication in human bladder epithelial cells. BMC Infect. Dis. 2004, 4, 16, DOI: 10.1186/1471-2334-4-16. Hertle, R. The family of Serratia type pore forming toxins. Curr. Protein Pept. Sci. 2005, 6, 313–325. DOI: 10.2174/1389203054546370]. As a hospitals infection, Serratia are facultative pathogens able to cause nosocomial infections or infections in immunocompromised patients, and these properties are studied rather scarily. We are not medical scientists. Moreover, invasion of S.grimesii is shown to be rather low, with only about 10% of the cells being invaded either by the wild-type or recombinant bacteria when cultures cells are infected in vitro [Bozhokina et al., 2011]. Hospital infection with S. grimesii or S.proteamaculansis is rather low. This implies that production of the enzyme does not make the bacteria pathogenic but rather provides them with an opportunity to render pathogenic under specific conditions. Therefore the aim of our review paper is quite different from the medical aspects. We have found that proteases grimelysin and protealysin isolated from S.grimesii and S.proteamaculans specifically cleave actin at a site strongly involved in actin functional activity and is not cleaved by any other protease. As cellular actin cytoskeleton is actively involved in the mechanisms of bacterial invasion, we believe that grimelysin and protealysin can play an important role in this process. As these processes are known rather poorly, our paper starts with the “actinase” properties of these new proteases probably important for the invasion mechanisms rather that to start with the bacterial invasion. We have modified Introduction and Conclusions of the paper to make them more clear.

I understand this is not a medical review and the mechanism of invasion in the case of S. grimesii and S. proteamaculansis is still rather unknown. However, I think that at least basic information about these bacteria species would be valuable for this paper. The authors are co-authors of the majority of all papers concerning grimelysin form Serratia, but other researchers do not necessarily know that these bacterial species are poorly characterized. Just for example the modified information that the Authors provided in answers could be incorporated in the text- “Moreover, invasion of S.grimesii is shown to be rather low, with only about 10% of the cells being invaded either by the wild-type or recombinant bacteria when cultures cells are infected in vitro [Bozhokina et al., 2011]. Hospital infection with S. grimesii or S.proteamaculansis is rather low.”

In respond to these comments, we have incorporated additional information (marked in red).

1. Conclusions. This part of the text begins now with an added paragraph concerning bacteria S. grimesii and S. proteamaculans:      Though Serratia are facultative pathogens able to cause nosocomial infections or infections in immunocompromised patients, hospital infection by S. grimesii or S. proteamaculans is low [64-66]. Consistent with this, the capability of S. grimesii to invade eukaryotic cells is also rather low, with only about 10% of the cells being invaded either by the wild-type or recombinant bacteria when cultured cells are infected in vitro [21}. This implies that production of grimelysin or protealysin does not make the bacteria pathogenic but rather provides them with an opportunity to render pathogenic under specific conditions. Therefore Serratia grimesii and Serratia proteamaculans invasion combines specific bacterial mechanisms to contact eukaryotic cells and specificity of the cells to be invaded.

  1. lines: 58-61- the sentence is still incomprehensible: “The same specific actin-hydrolyzing activity, characteristic of protease ECP 32 [9,10], was also revealed in bacterial extracts of both the reference S. grimesii strain and the corresponding recombinant protease from S. grimesii A2.” If the activity was presented in bacterial extracts of both strains then “the sentence and the corresponding recombinant protease from” should be deleted. Unless the Authors meant that in the bacterial extract from S. grimesii and in the protease S. grimessi A2, then the word both is misleading.

lines 59-61: The same specific actin-hydrolyzing activity, characteristic of protease ECP 32 [9,10], was also revealed in bacterial extracts of the reference S. grimesii strain and Escherichia coli transformed with the presumptive gene encoding grimelysin (ECP32) in S. grimesii A2 [18].

We are grateful to the reviewer for the helpful comments.
